# Efficacy of Biological Control Agents and Resistance Inducer for Control of Mal Secco Disease

**DOI:** 10.3390/plants12091735

**Published:** 2023-04-22

**Authors:** Giuseppa Rosaria Leonardi, Giancarlo Polizzi, Alessandro Vitale, Dalia Aiello

**Affiliations:** Dipartimento di Agricoltura, Alimentazione e Ambiente (Di3A), University of Catania, Via S. Sofia 100, 95123 Catania, Italy; giuseppa.leonardi@phd.unict.it (G.R.L.);

**Keywords:** disease management, acibenzolar-S-methyl, *Bacillus amyloliquefaciens*, *Trichoderma* spp., *Pythium oligandrum*

## Abstract

Mal secco, caused by *Plenodomus tracheiphilus,* is an economically important fungal vascular disease in citrus-growing countries of the Mediterranean basin. Preventing fungal infections usually requires a high number of copper treatments but European legislation imposes the minimization of their accumulation in soil. In our study, biological control agents (BCAs) and a plant resistance inducer (PRI), tested in four different experiments on citrus seedlings under controlled conditions, have resulted in promising strategies to control mal secco disease. Foliar (Experiment I) and soil (Experiment II) applications of two formulations of *Bacillus amyloliquefaciens* strain D747 (Amylo-X^®^ LC and Amylo-X^®^ WG) provided similar performances in reducing the disease amount (incidence and symptoms severity) over time compared to the untreated control, whereas copper hydroxide (Kocide Opti^®^) used as standard was the most effective treatment over time. In the third experiment, *Pythium oligandrum* strain M1 (Polyversum^®^) and *Trichoderma asperellum* strain ICC012 *+ Trichoderma gamsii* strain ICC080 (Remedier^®^) were able to reduce disease incidence and symptoms severity compared to the untreated control. Remedier^®^ provided the best performances in reducing the disease amount, whereas the Polyversum^®^ application was the least effective treatment over time. The effectiveness of the *Trichoderma* spp. formulation in reducing *P. tracheiphilus* infections did not significantly differ from the standard copper compound (Kocide Opti^®^). Comprehensively, in the last experiment (IV), acibenzolar-S-methyl (ASM) alone and in mixture with metalaxyl-M proved as effective as *B. amyloliquefaciens* strain FZB24, with no dose–response relationships observed. These findings provide important insight for the integrated management of mal secco disease.

## 1. Introduction

Mal secco, caused by the mitosporic fungus *Plenodomus tracheiphilus* (Petri) Gruyter, Aveskamp & Verkley (syn. *Phoma tracheiphila*), is the most destructive tracheomycotic disease endemically widespread in citrus-growing areas of the Mediterranean basin [1]. Although the pathogen mainly affects lemon (*C. limon* (L.) Burm. f.), other plant species may be infected, including bergamot (*C. bergamia* Risso), citron (*C. medica* L.), lime (*C. aurantifolia* Christ.), sour orange (*C. aurantium* L.), and volkamerian lemon (*C. volkameriana* Ten. & Pasq.) [2]. The pathogen enters through wounds, colonizes the xylematic vessels, and then spreads systemically, causing different syndromes according to the primary site of infection. Generally, symptoms start from the canopy as leaf, vein, and shoot chlorosis, which slowly evolves in the defoliation, wilt, and dieback of twigs and branches, and pathogens eventually lead to the plant’s death [3,4,5]. Control strategies, such as pruning of symptomatic twigs, eradication of severely infected plants, and preventive application of copper fungicides, have been adopted to reduce production losses. Despite this, an increase in disease severity has been reported in Sicily, probably due to a greater diffusion of the most susceptible lemon cultivar ‘’Femminello’’ and to hailfall damage, which is more and more frequent due to climate changes. In addition to cultural practices and the application of copper as a wound protectant, which is potentially subject to further restriction by the European Union [6], no other control measures are available to reduce the impact of mal secco disease. Currently, the main challenge faced by many researchers is turning more attention toward environmentally friendly and safe strategies to control plant diseases, including beneficial microorganisms and resistance inducers. Antagonistic bacteria, fungi, and oomycetes have been widely studied due to their good antifungal activity against a broad range of plant pathogens. Sporadic research efforts have been addressed to select biological control agents (BCAs) able to reduce infections of *P. tracheiphilus*. Although several endophytic bacteria, such as *Pseudomonas* spp. and *Bacillus* spp. [7,8,9,10], have shown promising results against mal secco, none of these have been screened to determine their impact on human health nor tested in field conditions to evaluate the potential application as trade available formulation. Assessment of commercial products represents a practical possibility often adopted to quickly find effective strategies and expedite procedures to enlarge the field use of a biocontrol agent. Among the BCAs deployed against plant pathogens, generalist microorganisms, such as *Trichoderma* and *Bacillus* species, are the most effective and commonly used for their survival ability under different abiotic and biotic changes as well as when encountering other microbes. However, there are few data in the literature discussing the use of BCAs commercially available to control mal secco. In our previous study, we investigated the role of four trade formulates of *Bacillus* strains against *P. tracheiphilus* through in vitro and in vivo experiments [11]. According to our findings in the in vitro tests, all bacterial strains were successful in suppressing mycelial growth in the majority of the *P. tracheiphilus* isolates studied. Likewise, all BCAs showed a biocontrol effectiveness in terms of decreasing disease values in *C. volkameriana* seedlings, supporting the findings for *Bacillus* species reported in prior research on mal secco disease [7,10]. However, no attempt was made to examine the association between efficacy, formulation, and the application mode of BCAs. There is some evidence that formulations may improve or reduce antagonistic efficacy, depending on the concentration of the bioproduct. The definition of the best application mode is a crucial step to enhancing the antagonistic activity of the studied biocontrol agents [12]. Moreover, to our knowledge no studies have been performed using other BCAs, such as fungal and oomycetes antagonists. *Trichoderma* spp. and *Pythium oligandrum* are known for their beneficial properties, such as their ability to interact with plants and other microorganisms and the production of antimicrobial and biostimulating metabolites, which allow their use in plant protection [13,14]. Recently, a considerable amount of literature has built up around plant resistance inducers (PRIs) to enhance defense mechanisms. Systemic acquired resistance (SAR), defined as an inducible response occurring from the localized infection of a pathogen or from treatment with elicitors, is an innate plant defense that may provide long-lasting protection against a wide range of pathogens. SAR is dependent on the signaling molecule salicylic acid (SA), and it is linked to the build-up of pathogenesis-related (PR) proteins, which are thought to be a factor in resistance [15,16]. Acibenzolar-S-methyl [benzo-(1,2,3)-thiadiazole-7-carbothioic acid S-methyl ester] is among the most well-known commercially available chemical resistance inducers that interfere with the salicylic acid (SA) pathway [17,18,19,20,21]. Acibenzolar-S-methyl has already been preliminarily and roughly tested on citrus against *P. tracheiphilus* [9], showing a reduction in mal secco symptoms. Therefore, the aims of this study were (i) to determine the influence of two formulations of *Bacillus amyloliquefaciens* strain D747, as well as different application modes, against *P. tracheiphilus*; (ii) to assess the effectiveness of two commercial products based on *Trichoderma* spp. and *Pythium oligandrum* in reducing *P. tracheiphilus* infections; and (iii) to compare the effectiveness between experimental formulates containing BCAs and a PRI against *P. tracheiphilus*.

## 2. Results

### 2.1. In Vivo Comparative Evaluation of Bacillus amyloliquefaciens Formulations and Mode of Application (Experiment I)

In Experiment I, where the bioformulates were sprayed onto the leaf, significant effects of single factors, treatment and time, were always detected, whereas the treatment × time interaction on disease incidence (DI) and symptoms severity (SS) was not significant (*p* > 0.01) (Table 1).

Following ANOVA, post-hoc analyses performed for each monitoring time clearly showed that all spray treatments were always significantly effective in reducing vein chlorosis caused by *P. tracheiphilus* on *C. volkameriana.* In detail, both Amylo-X^®^ WG and Amylo-X*^®^* LC provided similar performances in reducing significantly the disease amount (both DI and SS values), whereas the standard treatment, i.e., the Kocide Opti^®^ application, was the most effective treatment over time (Table 2).

Figure 1 shows the reduction in DI on seedlings treated with Amylo-X formulations, which was 55% (Day 14) and 49% (Day 28) for Amylo-X^®^ WG and 51% (Day 14) and 48% (Day 28) for Amylo-X^®^ LC, when compared with values obtained in the untreated control over time, whereas SS reduction was 60% (Day 14) and 53% (Day 28) for Amylo-X^®^ WG and 58% (Day 14) and 52% (Day 28) for Amylo-X^®^ LC. On the other hand, the reduction in the DI and SS of the copper hydroxide treatment 14 days after pathogen inoculation was 73% and 78%, respectively. With regard to disease progression over time, the disease amount data were higher in the second data point (Day 28) than in the first (Day 14), reporting an increase in DI values of 11%, 5%, and 14% for Amylo-X^®^ WG, Amylo-X^®^ LC, and copper hydroxide (Kocide Opti^®^), respectively, whereas there was an increase in SS of 11%, 14%, and 17% for Amylo-X^®^ WG, Amylo-X^®^ LC, and copper hydroxide (Kocide Opti^®^), respectively.

### 2.2. In Vivo Comparative Evaluation of Bacillus amyloliquefaciens Formulations and Mode of Application (Experiment II)

Similar to the previous experiment, in Experiment II the bioformulates were applied as root drench effects of single factors, i.e., treatment and time, on disease incidence and symptoms severity and were always significant unlike the treatment × time interactions, which were not significant (Table 3).

Post-hoc analyses performed for each monitoring time clearly showed that all root drench treatments were significantly effective in reducing vein chlorosis caused by *P. tracheiphilus* on *C. volkameriana*. The effectiveness ranking was the same as the previous experiment. Once again Kocide Opti^®^ provided the best performance over time in reducing the disease amount (both DI and SS values) followed by both Amylo-X^®^ WG and Amylo-X^®^ LC soil applications (Table 4).

Root applications of *B. amyloliquefaciens* D747 formulations determined a reduction in DI of 54% (Day 14) and 42% (Day 28) for Amylo-X^®^ WG and 56% (Day 14) and 45% (Day 28) for Amylo-X^®^ LC, when compared with values obtained in the untreated control over time, whereas SS reduction was 61% (Day 14) and 45% (Day 28) for Amylo-X^®^ WG and 62% (Day 14) and 48% (Day 28) for Amylo-X^®^ LC. On the other hand, the reduction in the DI and SS of the copper application 14 days after inoculation was 68% and 78%, respectively. With regard to disease progression over time, also in this experiment the disease amount data were higher in the second data point (Day 28) than in the first (Day 14), reporting an increase in DI values of 23%, 19%, and 14% and in SS values of 24%, 25%, and 19% for the Amylo-X^®^ WG, Amylo-X^®^ LC, and copper hydroxide applications, respectively (Table 4 and Figure 2).

### 2.3. In Vivo Biocontrol Activity of Trichoderma spp. and Pythium oligandrum Formulates (Experiment III)

In Experiment III, the effects of the single factors, i.e., treatment and time, were always significant, whereas the treatment × time interactions on disease incidence (DI) and symptoms severity (SS) were not significant (Table 5).

Following ANOVA, post-hoc analyses performed for each monitoring time clearly showed that all treatments were always significantly effective in reducing leaf spot caused by *P. tracheiphilus* on *C. volkameriana*. In detail, both Remedier^®^ and Kocide Opti^®^ provided the best performances in reducing the disease amount (both DI and SS values), whereas the Polyversum application was the least effective treatment over time (Table 6).

BCA treatments significantly reduced the disease amounts compared to the untreated control. *Pythium oligandrum* strain M1 showed a DI reduction of 41% (Day 14) and 28% (Day 28), whereas SS reduction was 36% (Day 14) and 34% (Day 28). *Trichoderma asperellum* strain ICC 012 + *T. gamsii* strain ICC 080 showed DI reductions of 61% (Day 14) and 49% (Day 28), whereas SS reduction was 66% (Day 14) and 56% (Day 28). Chemical applications resulted in the highest disease reduction, which ranged from 62% to 66% (DI) and from 65% to 77% (SS) 14 days after pathogen inoculation. Disease progression over time was observed, reporting disease amounts higher in the second data point (Day 28) than in the first (Day 14), with an increase in DI of 31%, 20%, and 6% and in SS of 7%, 15%, and 16% for *P. oligandrum* strain M1 (Polyversum^®^), *T. asperellum* strain ICC 012 + *T. gamsii* strain ICC 080 (Remedier^®^), and copper hydroxide (Kocide Opti^®^) applications, respectively (Table 6 and Figure 3).

### 2.4. Effects of Acibenzolar-S-Methyl in Reducing Mal Secco Disease (Experiment IV)

In Experiment IV (repeated in two trials) the performance of acibenzolar-S-methyl alone and combined with metalaxyl-M at various rates were compared with a bioformulate. The effects of treatment, trial, and treatment × trial interactions on disease parameter time were significant 14 days after inoculation, whereas the treatment × trial interactions were not significant 28 days after inoculation (Table 7). For this reason, data were analyzed for each trial.

In the first in vivo trial, post-hoc analysis revealed that all treatments were effective in reducing infections caused by *P. tracheiphilus* Pt18 (Table 8). Based on these data, all fungicide mixtures, fungicide alone, and BCAs showed similar performances in reducing fungal infections.

In detail, for this first experiment, the detected DI reduction with acibenzolar-S-methyl (ASM) applications was 44% (Day 14) and 33% (Day 28), whereas SS reduction was 52% (Day 14) and 42% (Day 28) when compared with values obtained in the untreated controls over time. The mixture of ASM + metalaxyl-M determined a DI reduction ranging from 35% to 51% (Day 14) and from 33% to 40% (Day 28), whereas SS reductions ranged from 41% to 59% (Day 14) and from 37% to 45% (Day 28). BCA application*s* were as effective as ASM in disease amount reduction, with DI reduction of 42–43% and SS reduction ranging from 32% to 52% over time (Table 8 and Figure 4). With regard to disease progression over time, SS data increased from 7% to 30% after 28 days when compared to those detected after 14 days (Table 8).

In the second trial of Experiment IV, post-hoc analysis revealed that all treatments were once again effective in reducing infections caused by *P. tracheiphilus* Pt18 (Table 9). Based on present data, all fungicide mixtures, fungicide alone, and BCAs showed similar performances in reducing fungal infections, although with a lower disease pressure thus confirming the data of the previous experiment.

In this experiment, the detected DI reduction with acibenzolar-S-methyl (ASM) applications was 29% (Day 14) and 35% (Day 28), whereas SS reduction was 38% (Day 14) and 47% (Day 28) when compared with values obtained in the untreated controls over time. The mixture of ASM + metalaxyl-M at three rates determined a DI reduction ranging from 40% to 47% (Day 14) and from 20% to 44% (Day 28), whereas SS reductions ranged from 45% to 55% (Day 14) and from 27% to 54% (Day 28). BCA applications were as effective as ASM in controlling disease, with DI reductions of 32% (Day 14) and 41% (Day 28) whereas SS reduction was 33% (Day 14) and 36% (Day 28) (Table 9 and Figure 5). With regard to disease progression over time, the DI and SS data of ASM + metalaxyl-M applications increased from 2% to 50% after 28 days when compared to those detected after 14 days (Table 9). Conversely, DI and SS reduction for ASM and *B. amyloliquefaciens* D747 applications was higher in the second data point (Day 28) than the first one (Day 14), with the reduction in disease amounts among treatments between 9% and 27%.

## 3. Discussion

Despite past efforts, the control of mal secco disease has remained largely unsuccessful. In our study, biocontrol agents and a resistant inducer were identified through in planta assays as promising viable means to introduce into mal secco management programs, where no alternatives to copper compounds are available.

Biocontrol approaches have been proposed, firstly focusing on *B. amyloliquefaciens* formulations. *Bacillus* is reported as one of the most important endophytic genera in citrus varieties, commonly used and commercialized as plant pathogen antagonists and plant growth promoters [22]. Starting from a previous study [11], we selected two commercial formulations of *B. amyloliquefaciens* strain D747, which has showed antifungal activity toward *P. tracheiphilus,* to investigate whether microbial loads and application methods could be involved in control efficacy. The findings showed no significant differences between the two formulations used, with both foliar and soil treatments effective in significantly decreasing the disease amount. Thus, the efficacy of *B. amyloliquefaciens* for the control of mal secco was not affected by the application method, and the strain D747 performed equally well using the two bioformulates with different microbial loads. However, the chemical was more effective than BCAs in controlling the disease, confirming the findings of our recent study [11]. The authors suggest that the ability of the bacterial strain to colonize systemically and to survive within the vascular tissue of citrus seedlings could be the most rational explanation for the good efficacy in reducing disease parameters even when using commercial products with a lower microbial load. This is supported by Kalai-Grami et al. [10], who demonstrated the xylem colonization ability of a strain of *B. amyloliquefaciens* isolated from citrus leaves, observing an antagonistic effect against *P. tracheiphilus* when pre-inoculated to the root zone. Likewise, in our previous study [11], *B. amyloliquefaciens* strains, including D747, were recovered from the stems of citrus seedlings after foliar application, confirming the ability to move and survive over time within the vascular system. Colonization ability, in fact, is considered a process required to perform the biocontrol functions, including the production of antimicrobial substances and lytic enzymes, nutrient and space competition, and induced systemic resistance (ISR) [22,23]. A deeper understanding of the colonization and survival of *Bacillus* strains under various environmental conditions is useful for increasing their use and effectiveness against *P. tracheiphilus*, for which selecting a potential biocontrol agent capable of colonizing the same ecological niche should increase the likelihood of effective control.

With regard to assessing the effectiveness of fungal- and oomycete-based bioformulates against *P. tracheiphilus,* our findings clearly revealed that *Trichoderma* spp. and *P. oligandrum* differ in their performances. In detail, the disease parameters were always significantly lower in seedlings sprayed with the fungal antagonists than in those treated with the oomycete. *Pythium oligandrum* is employed as a biocontrol agent against a wide range of fungal and oomycete plant pathogens, especially for its behavioral traits related to rhizosphere competence and mycoparasitism [13,24,25,26,27]. However, in the literature no data were found about its use for citrus diseases and particularly for mal secco disease. One unexpected finding was the strong reduction in symptoms observed in seedlings treated with *Trichoderma* spp., which did not differ from that determined by chemical application. This surprising result could bring enormous advantages to the management of mal secco, taking into account that none of the BCAs tested so far toward this pathogen performed as effectively as copper products [11]. The significant reduction in symptom development may suggest that the preventive activity of the application of BCA on wounds could represent a practical factor that influences biocontrol performance. According to recent reports [28,29,30,31], the protection effectiveness of *Trichoderma* spp. against plant pathogens is higher in preventive than in curative application when the time lapse between pathogen and antagonistic inoculation is limited. Generally, the application of BCAs before the pathogen allows the colonization of plant tissues and provides the time needed to activate the plant’s defense responses. In the present study, seedlings were treated with the antagonists twice with a 7-day interval before pathogen inoculation. Although in a preliminary way, our findings suggested that it could be a suitable time to ensure good efficacy. Up to now, what we knew about the fungal activity of *Trichoderma* toward *P. tracheiphilus* was largely based on a preliminary and outdated study performed to evaluate the in vitro antagonist activity of *Trichoderma* isolates from citrus soil [32]. Dual culture outcomes showed that *Trichoderma* spp. was able to overgrow on the *P. tracheiphilus* colony. Other interesting studies [33,34] were carried out to test transgenic lemon clones expressing chit42 gene of *T. harzianum*, showing that leaf proteins extracted from transgenic clones were able to inhibit the conidial germination and mycelial growth of *P. tracheiphilus*. Likewise, a susceptibility reduction was observed in transgenic plants after inoculation of the pathogen. Although transgenic plant cultivation is not currently authorized in Europe, these findings give an indication of a further potential role that *Trichoderma* spp. may have in the management of mal secco. As mentioned in the literature review, we must consider that *Trichoderma* species have different mechanisms of action and that additional genes besides chit42 are involved in biocontrol [35]. It is noteworthy to underline that *Trichoderma* species are able to interact with pathogenic targets through the production of other chemicals, such as antibiotics and lytic enzymes which have a role in the competition for nutrients and space, and in mycoparasitism. Likewise, they establish a relationship with the plant, changing its metabolism and enhancing defense responses to biotic and abiotic stresses by the production of elicitors. However, this molecular crosstalk communication greatly depends on the plant that antagonists are colonizing and the microorganisms that they are interacting with [36,37]. At present, more than 30 registered *Trichoderma* spp. products are available in Italy, but only one recorded on citrus is used to control fungal pathogens [38]. All the above-mentioned *Trichoderma* spp. applications are also largely used against pre- and post-harvest fungal diseases of citrus [30,39,40,41]. Nevertheless, to the authors’ knowledge this is the first report of an evaluation of *Trichoderma* spp. performance against mal secco disease under controlled conditions. Despite these promising data, to develop a full picture of the potential application of *Trichoderma*, further work is required to establish how many variables, e.g., species and strain, formulation, rate, time, and mode of application, may influence the effectiveness of *Trichoderma* in the sustainable management of mal secco.

Acibenzolar-S-methyl is an analog of the salicylic acid able to induce the systemic acquired resistance (SAR) through its metabolic pathway. With regard to the resistance inducer, alone and in combination with metalaxyl-M, and *B. amyloliquefaciens* strain FZB24, the performance in reducing the disease amount were similar. Overall, our data clearly indicated that no dose–response relationships existed between the concentrations of the mixture and its performances. It seems possible that this result is due to the mode of action of acibenzolar-S-methyl. Generally, several authors [42,43] have reported that the defense responses of the plant would most likely be triggered by a certain concentration of the active ingredient, above which the response is not enhanced. Moreover, it is noteworthy that no significant differences in reducing disease was observed between the mixture and acibenzolar-S-methyl alone, suggesting that metalaxyl-M is not effective against *P. tracheiphilus*. Indeed, metalaxyl-M is a phenylamide fungicide effective against oomycetes, acting on RNA polymerase I complex [44] and the development of a new fungicide containing acibenzolar-S-methyl and metalaxyl-M could allow the management of various citrus diseases.

## 4. Materials and Methods

### 4.1. Collection and Spore Preparation of Plenodomus tracheiphilus Isolates

*Plenodomus tracheiphilus* was isolated from lemon twigs with symptoms of salmon-pink wood discoloration. Diseased tissue fragments were surface disinfected for 1 min in 1% NaOCl, rinsed in sterile distilled water (SDW), and placed on potato dextrose agar (PDA, Lickson, Vicari, Italy) prepared by adding 100 mg L^−1^ of streptomycin sulfate (Sigma-Aldrich, St. Louis, MO, USA) to prevent bacterial growth. After incubation at 24 ± 1 °C under dark conditions for 10 days, single conidia were selected from culture plates to obtain single-spore isolates, which were placed in the fungi collection of the Dipartimento di Agricoltura, Alimentazione e Ambiente, Sezione di Patologia Vegetale, University of Catania. Identification of *P. tracheiphilus* was based on cultural morphology and microscopic observation as described by EPPO Bulletin [45]. The spore suspensions of isolates (Pt18, Pt41) used in the following assays were obtained from 20-day-old cultures maintained on PDA medium and stored at 22 °C. Thus, an aliquot of sterile water was added to the Petri plates and the mycelia were gently rubbed with a sterile loop, filtered through a triple layer of cheesecloth, and adjusted to a final concentration of 1 × 10^5^ conidia mL^−1^ using a microscope slide hemocytometer.

### 4.2. Plant Material and Growth Conditions

Four separate experiments were conducted using seedlings of volkamerian lemon (*C. volkameriana*), previously grown in a nursey greenhouse located in Giarre (Catania province, Italy) and maintained in plastic seed trays with a size of 540 × 280 mm and a depth of 110 mm. The plant material was obtained by healthy seed sown on a commercial substrate (90% blond peat + 10% perlite) previously fertilized with organic fertilizer (35–40% potassium nitrate, 0.3–1% copper sulfate penta-hydrate, 0.1–0.2% boric acid, 0.1–0.2% zinc sulfate) at 1 kg m^−3^ and with micro-nutrient fertilizer (15% Fe, 2.5% Mn, 0.20% B, 1% Cu, 1% Zn, 0.04% Mo) at 300 g m^−3^. Three weeks before the experiments, seedling trays were transferred to the growth chamber of the Dipartimento di Agricoltura, Alimentazione e Ambiente, University of Catania and maintained at 25 °C ± 1 °C, 80% of relative humidity, and in a 14 h light period.

### 4.3. In Vivo Comparative Evaluation of Bacillus amyloliquefaciens Formulations and Mode of Application (Experiments I, II)

To investigate the effect of formulation and application mode on disease control, a bacterial strain was chosen among the BCAs that had previously shown potential antifungal activity against *P. tracheiphilus* [11]. Two commercial formulations, Amylo-X^®^ WG (25% active ingredient (a.i.), water-dispersible granules containing 5 × 10^10^ CFU g^−1^) and Amylo-X^®^ LC (5% a.i., liquid concentration at 1 × 10^10^ CFU g^−1^), containing *Bacillus amyloliquefaciens* subsp. *plantarum* strain D747 and distributed by Biogard, CBC Europe S.r.l. (Grassobbio, Italy), were chosen and tested in different experiments using two modes of application: foliar spraying (Experiment I) and soil application (Experiment II). Three-month-old seedlings of *C. volkameriana* were treated twice (Experiment I) and three times (Experiment II) one week apart prior to the pathogen inoculation by spraying a volume of 100–150 mL of each formulate onto the leaves with a sprayer (Experiment I) and by manually distributing a volume of 30 mL to the soil of each plant pot (Experiment II). The doses used were 350 mL hL^−1^ and 200 g hL^−1^ for Amylo-X*^®^* LC and Amylo-X*^®^* WG, respectively. Two hours after the last application, the leaves were wounded at two midveins with a sterile needle and inoculated with approximately 0.8 mL of conidial suspension of Pt41. A total of four replicates per treatment were used, each including 16 seedlings, and three leaves in healthy condition per seedling were chosen to inoculate the pathogen, with a total of 384 inoculation points per treatment. A chemical control, consisting of seedlings treated once with copper hydroxide (30% a.i., Kocide Opti^®^, Certis Europe Italia, Saronno, Italy) at the highest label rate, and a positive control, treated with tap water and inoculated with *P. tracheiphilus*, were included.

### 4.4. In Vivo Biocontrol Activity of Trichoderma spp. and Pythium oligandrum Formulates (Experiment III)

Two commercial products were selected for the performance assessment against *P. tracheiphilus* infections: Remedier^®^, containing 2% of *T. asperellum* strain ICC 012 and 2% of *T. gamsii* strain ICC 080, and Polyversum^®^, containing 17.50% of *Pythium oligandrum* strain M1 (Gowan Italia S.p.A., Faenza, Italy). This experiment also included a chemical control (copper hydroxide, Kocide Opti^®^) and a positive control (seedlings inoculated with the pathogen). Three-month-old citrus (*C. volkameriana*) seedlings were treated twice 7 days apart before the pathogen inoculation by spraying a volume of 100–150 mL of each formulate onto the leaves with a manual sprayer. The application rates used in this study were 167 g hL^−1^, 20 g hL^−1^, and 150 g hL^−1^ for Remedier^®^, Polyversum^®^, and Kocide Opti^®^, respectively. Two hours after the second application, the leaves were wounded at two midveins with a sterile needle and inoculated with approximately 0.8 mL of conidial suspension of Pt41. A total of four replicates per treatment were used, each including 15 seedlings, and three leaves in healthy condition per seedling were chosen to inoculate the pathogen, with a total of 360 inoculation points per treatment. The control consisted of seedlings treated with tap water and inoculated with the pathogen.

### 4.5. Disease-Controlling Effect of Acibenzolar-S-Methyl (Experiment IV)

Formulates containing BCA and resistance inducer alone and in mixture with a fungicide, were kindly supplied by Syngenta Italia S.p.A. to investigate their effectiveness toward *P. tracheiphilus* infections under controlled conditions: A9180A (50% acibenzolar-S-methyl, WG), A9522B (4% acibenzolar-S-methyl, 38.76% metalaxyl-M, WP), and A20570A (13% *B. amyloliquefaciens* strain FZB24, WG). The latter was previously evaluated in our earlier study [11] against *P. tracheiphilus*. Comparisons were made between the applications of acibenzolar-S-methyl at 20 g hL^−1^, acibenzolar-S-methyl + metalaxyl-M at 10, 20, and 30 g hL^−1^ and *B. amyloliquefaciens* strain FZB24 at 37 g hL^−1^. Eleven-month-old citrus (*C. volkameriana*) seedlings were treated twice 7 days apart before the pathogen inoculation by spraying a volume of 100–150 mL of each formulate onto the leaves with a manual sprayer. After one week, the second application was made, where leaves were wounded at two midveins with a sterile needle and inoculated with approximately 0.8 mL of conidial suspension of Pt18. Each treatment was replicated three times with 15 seedlings used as one replicate. Four leaves in healthy condition per seedling were chosen to inoculate the pathogen, with a total of 360 inoculation points. Seven days later, two additional applications were performed with all formulates using volumes and methods previously described. The control consisted of seedlings treated with tap water and inoculated with the pathogen. The experiment was performed twice.

### 4.6. Disease Assessment

A total of 14 and 28 days after the pathogen inoculation, DI and SS parameters were evaluated. The DI value referred to the assessment of the percentage of positive inoculation points, whereas the SS value was counted on each inoculation point, adopting the empirical 0-to-4 rating scale of Luisi et al. [46], where: 0, no symptom; 1, chlorotic halo around the inoculation point; 2, vein chlorosis close to the inoculation point; 3, extended vein chlorosis to the leaf margin; 4, extensive vein chlorosis and/or browning close to the inoculation point.

Symptom severity was calculated using the following formula:$$\mathrm{SS}=\frac{\sum_{i=0}^{n}\left(\mathrm{Ci}\;\times \;n\right)}{N}$$
where: SS is the average index of symptom severity; Ci is each class detected; n is the number of inoculation points in each class; i (0-to-4) are the numerical values of the classes; N is the total number of inoculation points examined.

### 4.7. Data Analysis

All the statistical analyses of in vivo data were carried out using the Statistica package software (version 10; Statsoft Inc., Tulsa, OK, USA). The average disease incidence (DI) and symptom severity (SS) data were calculated by averaging the values determined for the single replicates of each treatment data and reported in the tables. Percentage disease incidence data were previously arcsine (sin^−1^ square root x) transformed to meet the assumptions of homogeneity of variance. In the post-hoc analyses, the main effects of the treatments were evaluated by using one-way analysis of variance, and the means were separated by the Fisher’s least significance difference test (α = 0.05 in Experiments I, II, and III and α = 0.01 in Experiment IV). Moreover, the performance of the treatments on all disease parameters were also compared to the untreated control by using the Abbott’s formula.

## 5. Conclusions

The future for the management of mal secco disease is highly dependent on the technical placement of the few trade products that are currently available, taking into account those with low impact on the environment and human health compared with commonly used fungicides. The present paper shows that important measures are currently present for the integrated management of mal secco. Taken together, our findings, although preliminary, have observed a variable control level obtained among bacterial, fungal, and oomycete antagonists, indicating *Trichoderma* spp. as potent biocontrol means to manage mal secco. Although this work contributes to existing knowledge on mal secco control, there are some aspects to definitively resolve the BCAs–pathogen–host relationship about which little is known. Further research is an essential next step to confirm the effective use in the field of BCAs and the resistance inducer tested in this study.

## Figures and Tables

**Figure 1 plants-12-01735-f001:**
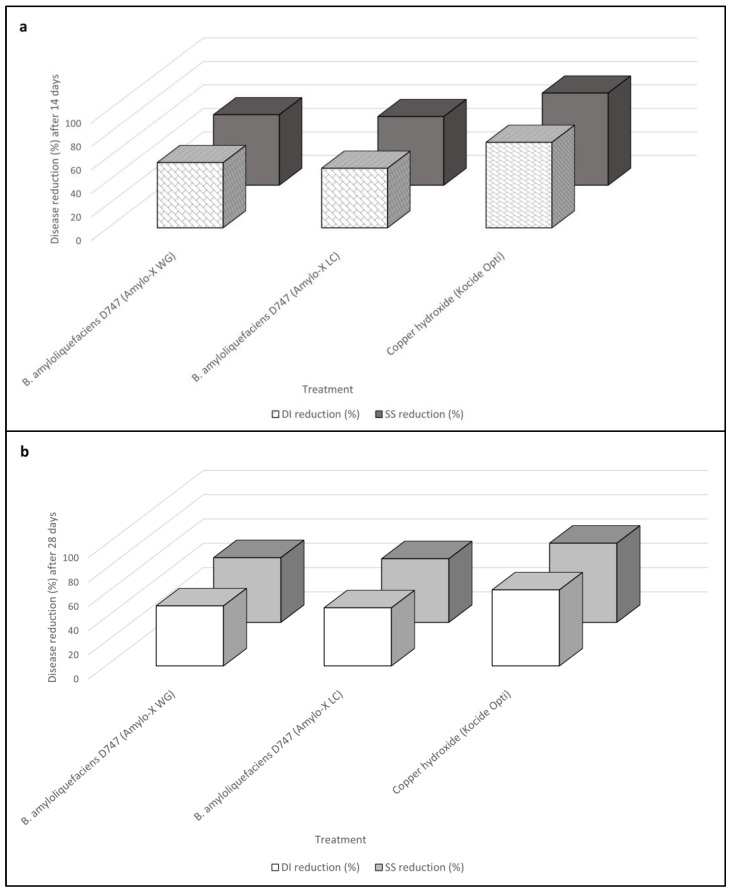
Reduction in disease incidence (DI) and symptoms severity (SS) according to Abbott’s indexes for *Bacillus amyloliquefaciens* strain D747 (Amylo-X^®^ LC and Amylo-X^®^ WG) and copper hydroxide (Kocide Opti^®^) foliar applications calculated over time, 14 (**a**) and 28 (**b**) days after the inoculation of *Plenodomus tracheiphilus*. Disease incidence (DI) and symptoms severity (SS) data are compared with each other.

**Figure 2 plants-12-01735-f002:**
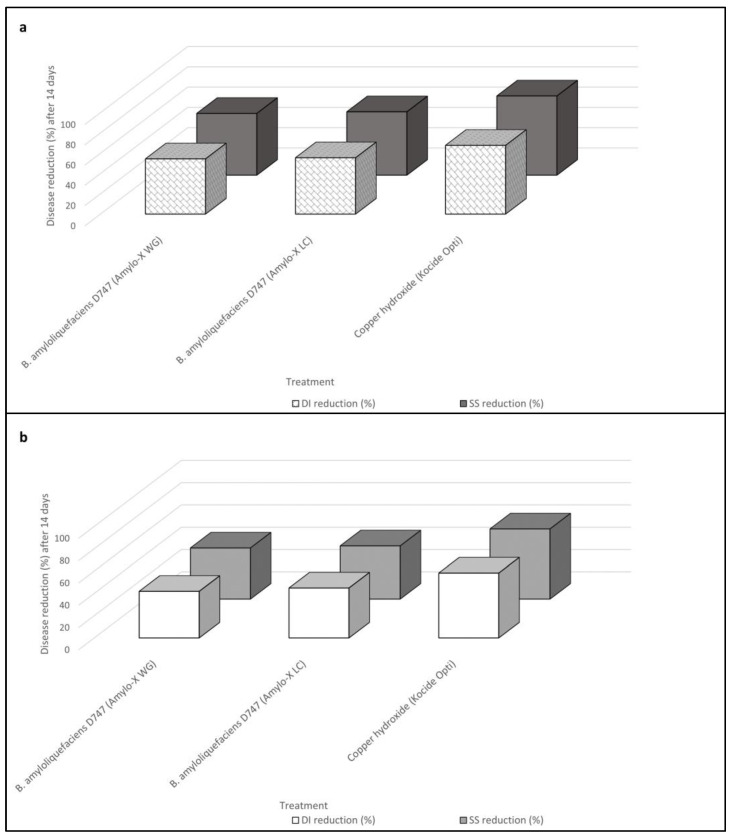
Reduction in disease incidence (DI) and symptoms severity (SS) according to Abbott’s indexes for *Bacillus amyloliquefaciens* strain D747 (Amylo-X^®^ WG and Amylo-X^®^ LC) soil applications and copper hydroxide (Kocide Opti^®^) applications calculated over time, 14 (**a**) and 28 (**b**) days after the inoculation of *Plenodomus tracheiphilus*. Disease incidence (DI) and severity symptoms (SS) data are compared with each other.

**Figure 3 plants-12-01735-f003:**
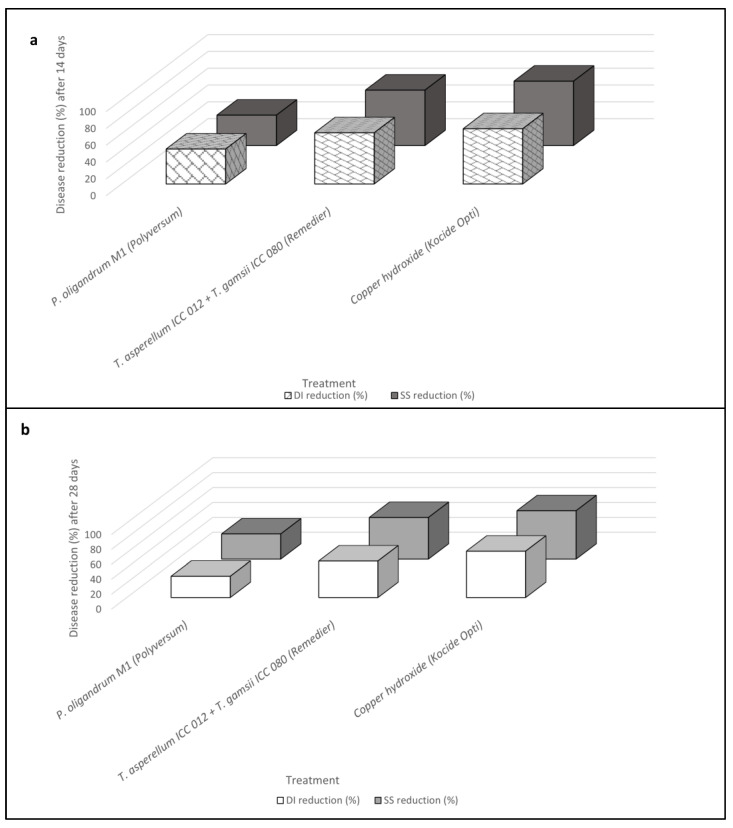
Reduction in disease incidence (DI) and symptoms severity (SS) according to Abbott’s indexes for *Pythium oligandrum* M1 (Polyversum^®^), *Trichoderma asperellum* ICC 012 + *Trichoderma gamsii* ICC 080 (Remedier^®^), and copper hydroxide (Kocide Opti^®^) applications calculated over time, 14 (**a**) and 28 (**b**) days after the inoculation of *Plenodomus tracheiphilus*. Disease incidence (DI) and severity symptoms (SS) data are compared with each other.

**Figure 4 plants-12-01735-f004:**
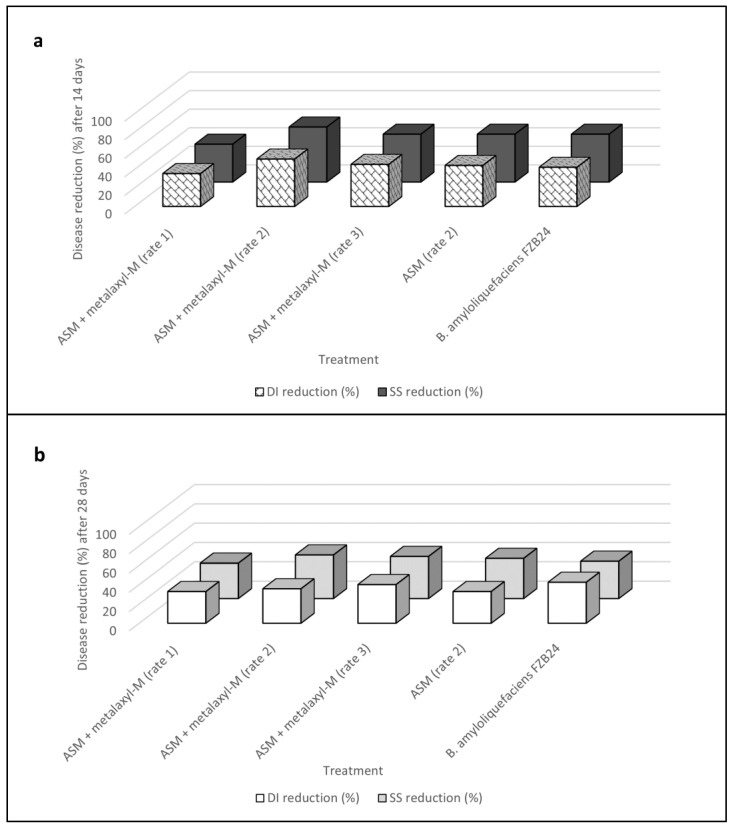
Reduction in disease incidence (DI) and symptoms severity (SS) according to Abbott’s indexes for acibenzolar-S-methyl (ASM) alone and in mixture with metalaxyl-M and *Bacillus amyloliquefaciens* strain FZB24 treatments calculated over time, 14 (**a**) and 28 (**b**) days after the inoculation of *Plenodomus tracheiphilus*, in the first trial of Experiment IV. Disease incidence (DI) and severity symptoms (SS) data are compared with each other.

**Figure 5 plants-12-01735-f005:**
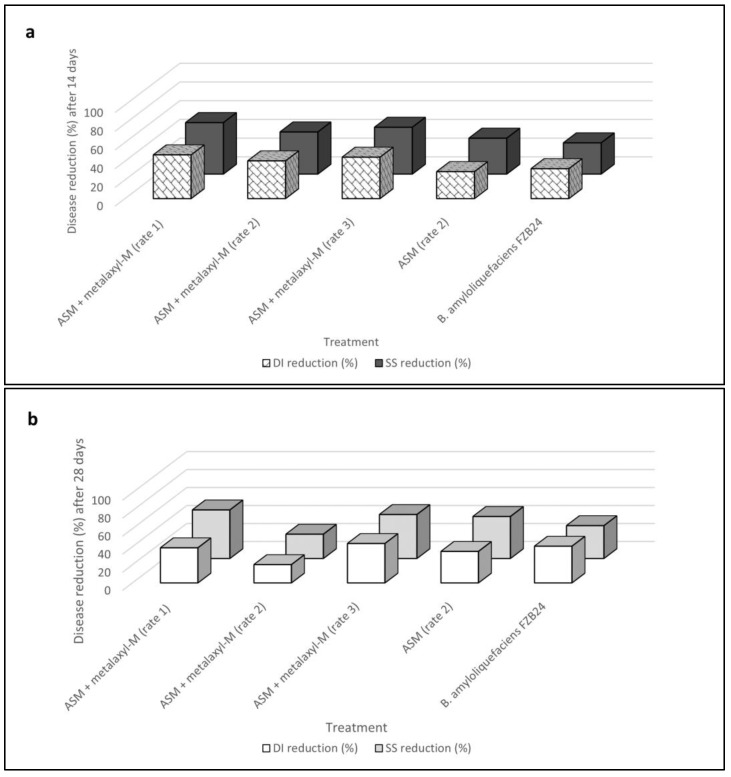
Reduction in disease incidence (DI) and symptoms severity (SS) according to Abbott’s indexes for acibenzolar-S-methyl (ASM) alone and in mixture with metalaxyl-M and *Bacillus amyloliquefaciens* strain FZB24 treatments calculated over time, 14 (**a**) and 28 (**b**) days after the inoculation of *Plenodomus tracheiphilus*, in the second trial of Experiment IV. Disease incidence (DI) and severity symptoms (SS) data are compared with each other.

**Table 1 plants-12-01735-t001:** Analysis of variance for disease incidence and symptom severity caused by *Plenodomus tracheiphilus* of leaf spray treatments over time on *Citrus volkameriana*.

Factor(s)		Disease Incidence ^y^		Disease Severity ^y^	
	df	*F*	*p Value*	*F*	*p Value*
Treatment	3	132.622	*>0.0000*	120.403	*>0.0000*
Time	1	13.215	*0.001317*	145.042	*>0.0000*
Treatment × time	3	0.395	0.757782 ^ns^	3.903	0.021058 ^ns^

^y^ F test of fixed effects; df = degrees of freedom; *p* value associated to F; ^ns^ = not significant data.

**Table 2 plants-12-01735-t002:** Post-hoc analysis on the main effects of leaf spray treatments in reducing disease incidence and symptom severity caused by *Plenodomus tracheiphilus* strain Pt41 on *Citrus volkameriana* 14 and 28 days after inoculation.

	After 14 Days			After 28 Days		
Treatments	DI (%) ^y^	DI (va) ^y^	SS (0-to-4) ^y^	DI (%) ^y^	DI (va) ^y^	SS (0-to-4) ^y^
Untreated control	72.50 ± 3.13 a	58.50	1.63 ± 0.08 a	77.75 ± 3.43 a	62.10	2.73 ± 0.13 a
*Bacillus amyloliquefaciens* strain D747 (Amylo-X*^®^* WG)	32.25 ± 1.75 b	34.57	0.65 ± 0.06 b	39.25 ± 3.08 b	38.75	1.28 ± 0.10 b
*Bacillus amyloliquefaciens* strain D747 (Amylo-X*^®^* LC)	35.75 ± 2.31 b	36.68	0.68 ± 0.10 b	40.10 ± 1.10 b	39.52	1.30 ± 0.08 b
Copper hydroxide (Kocide Opti*^®^*)	19.75 ± 1.43 c	26.34	0.35 ± 0.03 c	29.00 ± 1.13 c	32.57	0.95 ± 0.03 c

^y^ Data derived from four replicates each including 16 seedlings (six inoculation points/seedling). Standard error of the mean = SEM. Percentage disease incidence (DI) data were previously arcsine (sin^−1^ square root x) transformed to meet the assumptions of homogeneity of variance. Means followed by different letters within the column are significantly different according to Fisher’s least significant differences test (α = 0.05).

**Table 3 plants-12-01735-t003:** Analysis of variance for disease incidence and symptoms severity caused by *Plenodomus tracheiphilus* of root drench treatments over time on *Citrus volkameriana*.

Factor(s)		Disease Incidence ^y^		Disease Severity ^y^	
	df	*F*	*p Value*	*F*	*p Value*
Treatment	3	98.599	*>0.0000*	74.577	*>0.0000*
Time	1	28.786	*0.000017*	121.467	*>0.0000*
Treatment × time	3	0.258	0.854763 ^ns^	1.273	0.306233 ^ns^

^y^ F test of fixed effects; df = degrees of freedom; *p* value associated to F; ^ns^ = not significant data.

**Table 4 plants-12-01735-t004:** Post-hoc analysis on main effects of root drench treatments in reducing disease incidence and symptoms severity caused by *Plenodomus tracheiphilus* strain Pt41 on *Citrus volkameriana* 14 and 28 days after inoculation.

	After 14 Days			After 28 Days		
Treatments	DI (%) ^y^	DI (va) ^y^	SS (0-to-4) ^y^	DI (%) ^y^	DI (va) ^y^	SS (0-to-4) ^y^
Untreated control	67.50 ± 2.74 a	55.30	1.60 ± 0.08 a	74.75 ± 3.99 a	60.04	2.63 ± 0.20 a
*Bacillus amyloliquefaciens* strain D747 (Amylo-X*^®^* WG)	30.75 ± 1.53 b	33.65	0.63 ± 0.07 b	43.50 ± 1.71 b	41.26	1.43 ± 0.08 b
*Bacillus amyloliquefaciens* strain D747 (Amylo-X*^®^* LC)	30.00 ± 1.51 b	33.19	0.60 ± 0.04 b	41.25 ± 2.86 b	39.94	1.38 ± 0.08 b
Copper hydroxide (Kocide Opti*^®^*)	21.75 ± 2.46 c	27.67	0.35 ± 0.06 c	31.25 ± 1.90 c	33.95	0.98 ± 0.06 c

^y^ Data derived from four replicates each including 16 seedlings (six inoculation points/seedling). Standard error of the mean = SEM. Percentage disease incidence (DI) data were previously arcsine (sin^−1^ square root x) transformed to meet the assumptions of homogeneity of variance. Means followed by different letters within the column are significantly different according to Fisher’s least significant differences test (α = 0.05).

**Table 5 plants-12-01735-t005:** Analysis of variance for disease incidence and severity of leaf spot caused by *Plenodomus tracheiphilus* treatments over time on *Citrus volkameriana*.

Factor(s)		Disease Incidence ^y^		Disease Severity ^y^	
	df	*F*	*p Value*	*F*	*p Value*
Treatment	3	58.254	*>0.0000*	59.9153	*>0.0000*
Time	1	10.275	*0.003790*	73.4746	*>0.0000*
Treatment × time	3	0.619	0.60933 ^ns^	1.5559	0.225941 ^ns^

^y^ F test of fixed effects; df = degrees of freedom; *p* value associated to F; ^ns^ = not significant data.

**Table 6 plants-12-01735-t006:** Post-hoc analysis on the main effects of treatments in reducing leaf spot caused by *Plenodomus tracheiphilus* strain Pt41 on *Citrus volkameriana* fourteen and twenty-eight days after inoculation.

	After 14 Days			After 28 Days		
*Treatments*	DI (%) ^y^	DI (va) ^y^	SS (0-to-4) ^y^	DI (%) ^y^	DI (va) ^y^	SS (0-to-4) ^y^
Untreated control	51.75 ± 2.34 a	46.01	1.18 ± 0.06 a	55.25 ± 2.57 a	48.03	1.93 ± 0.07 a
*Pythium oligandrum* M1 (Polyversum*^®^*)	30.25 ± 3.53 b	33.21	0.75 ± 0.15 b	39.50 ± 2.84 b	38.90	1.28 ± 0.10 b
*Trichoderma asperellum* ICC 012 + *T. gamsii* ICC 080 (Remedier*^®^*)	20.25 ± 2.15 c	26.64	0.40 ± 0.04 c	28.25 ± 2.66 c	32.02	0.85 ± 0.08 c
Copper hydroxide (Kocide Opti*^®^*)	17.75 ± 2.12 c	24.79	0.28 ± 0.04 c	21.00 ± 0.76 c	27.26	0.68 ± 0.02 c

^y^ Data derived from four replicates each including 15 seedlings (six inoculation points/seedling). Standard error of the mean = SEM. Percentage disease incidence (DI) data were previously arcsine (sin^−1^ square root x) transformed to meet the assumptions of homogeneity of variance. Means followed by different letters within the column are significantly different according to Fisher’s least significant differences test (α = 0.05).

**Table 7 plants-12-01735-t007:** Analysis of variance for disease incidence and severity symptoms caused by *Plenodomus tracheiphilus* of acibenzolar-S-methyl-based treatments over time applied on *Citrus volkameriana*.

		14 Days				28 Days			
Factor(s)		DI ^y^		SS ^y^		DI ^y^		SS ^y^	
	df	*F*	*p Value*	*F*	*p Value*	*F*	*p Value*	*F*	*p Value*
Treatment	5	28.791	*>0.0000*	27.428	*>0.0000*	*13.919*	*>0.0000*	*17.114*	*>0.0000*
Trial	1	93.793	*>0.0000*	154.793	*>0.0000*	*14.124*	*>0.0000*	*56.860*	*>0.0000*
Trt × trial	5	4.003	*0.0088*	6.903	*0.0004*	0.912	0.4896	1.974	0.1191

^y^ F test of fixed effects; df = degrees of freedom; *p* value associated to F.

**Table 8 plants-12-01735-t008:** Post-hoc analysis on the main effects of treatments in reducing infections of *Plenodomus tracheiphilus* strain Pt18 in the first trial of Experiment IV.

		14 Days			28 Days	
Treatments	DI (%) ^y^	DI (va) ^y^	SS (0-to-4) ^y^	DI (%) ^y^	DI (va) ^y^	SS (0-to-4) ^y^
Untreated control	64.0 ± 1.2 a	53.1	1.3 ± 0.06 a	69.0 ± 4.5 a	56.3	2.0 ± 0.1 a
Acibenzolar-S-methyl + metalaxyl-M (rate 1)	41.3 ± 4.2 b	40.0	0.8 ± 0.09 b	46.3 ± 4.4 b	42.9	1.3 ± 0.1 b
Acibenzolar-S-methyl + metalaxyl-M (rate 2)	31.3 ± 4.2 b	33.9	0.5 ± 0.03 b	44.3 ± 5.7 b	41.7	1.1 ± 0.1 b
Acibenzolar-S-methyl + metalaxyl-M (rate 3)	35.0 ± 1.0 b	36.3	0.6 ± 0.03 b	41.3 ± 1.4 b	40.0	1.1 ± 0.1 b
Acibenzolar-S-methyl	35.7 ± 1.7 b	36.7	0.6 ± 0.09 b	46.3 ± 4.4 b	42.9	1.2 ± 0.2 b
*Bacillus amyloliquefaciens* strain FZB24	37.0 ± 1.0 b	37.5	0.6 ± 0.03 b	39.7 ± 1.7 b	39.0	1.2 ± 0.1 b

^y^ Data derived from three replicates each including 15 seedlings (eight inoculation points/seedling). Standard error of the mean = SEM. Percentage disease incidence (DI) data were previously arcsine (sin^−1^ square root x) transformed to meet the assumptions of homogeneity of variance. Means followed by different letters within the column are significantly different according to Fisher’s least significant differences test (α = 0.01).

**Table 9 plants-12-01735-t009:** Post-hoc analysis on the main effects of treatments in reducing infections of *Plenodomus tracheiphilus* strain Pt18 in the second trial of Experiment IV.

		14 Days			28 Days	
Treatments	DI (%) ^y^	DI (va) ^y^	SS (0-to-4) ^y^	DI (%) ^y^	DI (va) ^y^	SS (0-to-4) ^y^
Untreated control	40.7 ± 1.4 a	39.6	0.6 ± 0.05 a	56.7 ± 2.3 a	48.8	1.4 ± 0.09 a
Acibenzolar-S-methyl + metalaxyl-M (rate 1)	21.7 ± 1.8 b	27.7	0.3 ± 0.03 b	34.7 ± 2.6 b	36.0	0.6 ± 0.09 b
Acibenzolar-S-methyl + metalaxyl-M (rate 2)	24.3 ± 2.0 b	29.5	0.3 ± 0.03 b	45.3 ± 1.7 b	42.3	1.0 ± 0.06 b
Acibenzolar-S-methyl + metalaxyl-M (rate 3)	22.7 ± 1.7 b	28.4	0.3 ± 0.00 b	32.0 ± 5.9 b	34.2	0.7 ± 0.1 b
Acibenzolar-S-methyl	29.0 ± 1.1 b	32.6	0.4 ± 0.03 b	37.0 ± 3.2 b	37.4	0.7 ± 0.1 b
*Bacillus amyloliquefaciens* strain FZB24	27.7 ± 2.9 b	31.7	0.4 ± 0.06 b	33.7 ± 1.4 b	35.5	0.9 ± 0.09 b

^y^ Data derived from three replicates each including 15 seedlings (eight inoculation points/seedling). Standard error of the mean = SEM. Percentage disease incidence (DI) data were previously arcsine (sin^−1^ square root x) transformed to meet the assumptions of homogeneity of variance. Means followed by different letters within the column are significantly different according to Fisher’s least significant differences test (α = 0.01).

## Data Availability

The data presented in this study are available on request from the corresponding author.

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
