# Peer review of "Efficacy of Biological Control Agents and Resistance Inducer for Control of Mal Secco Disease"

_plants, 2023, doi:10.3390/plants12091735_

Round 1
Reviewer 1 Report
Mal secco, a common citrus fungal disease in the Mediterranean region, used to be controlled by copper treatments. However, due to new restrictions imposed by European legislation, alternative strategies such as biological control agents (BCAs) are needed. In this manuscript, the authors 1) tested the efficacy of two candidate BCAs, Amylo-X® WG and Am-115 ylo-X® LC, and found that they were not as effective as the copper compound Kocide Opti®. 2) Tested the effect of different BCA application modes and found that application modes do not have a significant effect on BCA efficiency. 3) Tested two commercial BCAs, Polyversum® and Remedier®, and concluded that Remedier® is as effective as Kocide Opti®. 4) Resistance Inducer was also tested. Overall, the research is interesting and well presented. The only problems I noticed are the figures. I think the figures should be improved to make them easier to understand. What the figures show are "reductions" of DI and SS, but the legends just say "14/28 days SS/DI", which means the opposite and can be misleading. The way the SS and DI data are presented, in my opinion, adds to the difficulty of understanding. I suggest either to separate SS and DI and make them part A and B of figure x, or to put the SS and DI bars of the same days side by side.
Author Response
Q1: Mal secco, a common citrus fungal disease in the Mediterranean region, used to be controlled by copper treatments. However, due to new restrictions imposed by European legislation, alternative strategies such as biological control agents (BCAs) are needed. In this manuscript, the authors 1) tested the efficacy of two candidate BCAs, Amylo-X® WG and Amylo-X® LC and found that they were not as effective as the copper compound Kocide Opti®. 2) Tested the effect of different BCA application modes and found that application modes do not have a significant effect on BCA efficiency. 3) Tested two commercial BCAs, Polyversum® and Remedier®, and concluded that Remedier® is as effective as Kocide Opti®. 4) Resistance Inducer was also tested. Overall, the research is interesting and well presented.
A1: Thank you for the comment.
Q2: The only problems I noticed are the figures. I think the figures should be improved to make them easier to understand. What the figures show are "reductions" of DI and SS, but the legends just say "14/28 days SS/DI", which means the opposite and can be misleading. The way the SS and DI data are presented, in my opinion, adds to the difficulty of understanding. I suggest either to separate SS and DI and make them part A and B of figure x, or to put the SS and DI bars of the same days side by side.
A2: Thanks for suggestion. Done. The authors chose for the first option suggested by you.
Reviewer 2 Report
The conducted research is of great practical importance and is in line with the current trends in the protection of plants against fungal pathogens. This type of research, four different experiences regarding the form and number of applications, the variety of preparations, incl. based on copper compounds and exhibiting an antagonistic effect in relation to fungal pathogens are very right and needed for practice.
I have a few comments about the chapter: Materials and Methods
- please provide in all experiments the varieties of lemons used in the experiments
- in experiments 2 and 3, please complete information on three-month-old C. volkameriana seedlings, how they were propagated, whether the experiment was carried out on material in a nursery or after planting on a commercial plantation, please write something about the cultivation history if it was a newly planted plantation, what kind soil, growing region, etc.
- similarly, please complete the information on the plant material in experiment 4.
Author Response
Q1: The conducted research is of great practical importance and is in line with the current trends in the protection of plants against fungal pathogens. This type of research, four different experiences regarding the form and number of applications, the variety of preparations, incl. based on copper compounds and exhibiting an antagonistic effect in relation to fungal pathogens are very right and needed for practice.
A1: Thank you for the comment.
Q2: In experiments 2 and 3, please complete information on three-month-old C. volkameriana seedlings, how they were propagated, whether the experiment was carried out on material in a nursery or after planting on a commercial plantation, please write something about the cultivation history if it was a newly planted plantation, what kind soil, growing region, etc. Similarly, please complete the information on the plant material in experiment 4.
A2: Thank you for the comment. The authors modified the manuscript according to your suggestions.